# Photocatalytic Oxidative Desulfurization of Thiophene by Exploiting a Mesoporous V₂O₅-ZnO Nanocomposite as an Effective Photocatalyst

**Maha Alhaddad** [1,*], **Ahmed Shawky** [2,*] and **Zaki I. Zaki** [3]

[1] Department of Chemistry, Faculty of Science, King Abdulaziz University, P.O. Box 80203, Jeddah 21589, Saudi Arabia
[2] Nanomaterials and Nanotechnology Department, Advanced Materials Institute, Central Metallurgical Research and Development Institute, P.O. Box 87, Cairo, Helwan 11421, Egypt
[3] Department of Chemistry, College of Science, Taif University, P.O. Box 11099, Taif 21944, Saudi Arabia
* Correspondence: mahahaddad101@gmail.com (M.A.); phyashawky@gmail.com (A.S.)

**Abstract:** Due to increasingly stringent environmental regulations imposed by governments throughout the world, the manufacture of low-sulfur fuels has received considerable assiduity in the petroleum industry. In this investigation, mesoporous V₂O₅-decorated two-dimensional ZnO nanocrystals were manufactured using a simple surfactant-assisted sol–gel method for thiophene photocatalytic oxidative desulfurization (TPOD) at ambient temperature applying visible illumination. When correlated to pure ZnO NCs, V₂O₅-added ZnO nanocomposites dramatically improved the photocatalytic desulfurization of thiophene, and the reaction was shown to follow the pseudo-first-order model. The photocatalytic effectiveness of the 3.0 wt.% V₂O₅-ZnO photocatalyst was the greatest among all the other samples, with a rate constant of $0.0166$ min$^{-1}$, which was 30.7 significantly greater than that of pure ZnO NCs ($0.00054$ min$^{-1}$). Compared with ZnO NCs, and owing to their synergetic effects, substantial creation of hydroxyl radical levels, lesser light scattering action, quick transport of thiophene species to the active recenters, and efficient visible-light gathering, V₂O₅-ZnO nanocomposites were found to have enhanced photocatalytic efficiency. V₂O₅-ZnO nanocomposites demonstrated outstanding stability during TPOD. Using mesoporous V₂O₅-ZnO nanocomposites, the mechanism of the charge separation process was postulated.

**Keywords:** desulfurization; mesoporous; photocatalysis; thiophene; V₂O₅-ZnO; visible illumination

## 1. Introduction

Sulfate particles and exhaust gas are primarily formed from sulfur-containing fuels, which are the primary drivers of acid rain and pose a serious threat to human health and the ecosystems [1,2]. Recently, catalyzed hydrodesulfurization (HDS) is used in industry sectors to desulfurize fuels, with catalysts made of NiMo₂ and CoMo sulfides. Even though HDS requires harsh circumstances such as a high H₂ input, temperature, and pressure, it may transform sulfur-containing organic molecules such as thiophene into H₂S, which can be easily disposed of and segregated [1]. Because of its aromaticity and the low electron density of sulfur, thiophene-stimulated desulfurization demands a huge effort in oxidation; as a result, it is critical to design an active photocatalyst for desulfurization as well as a hybrid approach based on complete oxidation processes [2]. Selecting the most appropriate regime for catalyst synthesis for a particular application is an essential issue [3]. The nanostructured materials' special characteristics, such as efficacious electronic and optical features, high mechanical resistance, high refractive index, non-toxic nature, and excellent photostability, make them an excellent option for a multitude of purposes as efficient catalysts and photocatalysts [4–12]. As the reactivity of aromatic C-S bonds is substantially identical to that of oil hydrocarbons, it is hard to transform thiophene compounds into

$SO_x$ products using conventional moderate desulfurization procedures for the creation of ecologically friendly fuels. Rather, they are mostly converted into sulfoxides and sulfones that are difficult to separate from the fuel source [1]. Using hydrogen and a Ti-based catalyst, several catalytic desulfurization techniques for thiophene decomposition have been described, with a sulfur removal efficiency of 60–90% from fuel at high temperatures [13,14]. Alternative catalytic desulfurization research that used Fenton reagent and acetic acid to separate thiophene from simulated crude oil accomplished a sulfur elimination of nearly 70% after 30 min [15]. During the 3 h of crude oil desulfurization leading to an expanded surface area metallic framework, around 93–99% of the sulfur was eliminated [16]. Because of its many implications in the removal of pollutants, such as the removal of hazardous ions [17,18] and a wide range of organic pollutants [19–21], aside from its utility in the production of important chemical molecules [22,23], photocatalysis has gained considerable attention among researchers. A promising regime for eliminating sulfur-containing chemicals from the fuel is photocatalytic oxidative desulfurization [24,25]. For enhancing photocatalytic efficiency, active nanocrystals (NCs) with high dispersal rates and reduced sizes are beneficial [26,27]. They are dependent on catalyst supports' surface area, crystal phase, and microstructure [28,29]. As a result, the precise manufacturing of ZnO catalysts with suitable morphologies is critical and remains a significant barrier to promoting superior catalytic effectiveness for biomass improvement [30,31]. Transition metal oxides with mesoporous architectures are clearly in great demand owing to their unique properties such as mechanical stress buffering, light absorption, and expanded surface area, in addition to low density for target molecule diffusion and adsorption [32–36]. The photocatalytic activity of ZnO-based catalysts at the nanoscale is remarkable. Additionally, having a porous texture, in combination with a carefully regulated shape and size, can result in catalysts with unique functionalities for particular tasks. Using sacrificial silica species as a hard template, ZnO spheres can be created [23–36]. Regrettably, owing to the template's removal step disadvantage, the processes for synthesis employing hard templates are limited, and the calcination process may cause the porous structures to crumble [37]. As a result, developing a low-cost, straightforward technique for generating mesoporous ZnO with increased stability and an appropriate design represents a significant problem. A p-type $V_2O_5$ semiconductor, on the other hand, does have a low bandgap energy rate of about 2.2 eV [38]. Furthermore, $V_2O_5$ is commonly used to enhance visible light harvesting and promote photonic effectiveness. As a consequence, due to easy fabrication methods, operability, and high performance, it is appropriate to develop heterojunctions by coupling $V_2O_5$ with another metal oxide such as $V_2O_5$-$CeO_2$ [39], $V_2O_5$-$ZrO_2$ [40], or $V_2O_5$-$TiO_2$ [41]. The special features of $V_2O_5$ make it a perfect choice for manufacturing mesoporous $V_2O_5$-ZnO nanocomposites for extremely efficient photocatalysts using sunlight. Only a few studies have been conducted on the production of $V_2O_5$-ZnO nanocomposites and their potential use as effective photocatalysts for the desulfurization of sulfur-containing compounds. It is worth noting that the shape, morphology, surface area, and geometry of the catalysts all impact their photocatalytic activity [42–44]. Due to the ease of diffusion, as well as the adsorption of target species, high surface-to-volume ratio, and excellent incident light absorption, mesoporous nanocomposites are extensively employed as photocatalysts, according to various research works [45]. Mesoporous $V_2O_5$-ZnO nanocomposites were constructed in this study for the photocatalytic oxidative desulfurization of thiophene with visible illumination at room temperature. When compared with ZnO NCs, the photocatalytic desulfurization of thiophene was greatly boosted by $V_2O_5$-ZnO nanostructured materials, which fitted the pseudo-first-order rule. In comparison to the other tested materials, the photocatalytic efficacy of the 3.0 wt.% $V_2O_5$-ZnO photocatalyst was the largest, with a velocity constant value of 0.0166 min$^{-1}$, which was 30.7-fold that of ZnO NCs (0.00054 min$^{-1}$). Using mesoporous $V_2O_5$-ZnO nanocomposites, the mechanism of the charge separation process was postulated.

## 2. Results and Discussion

### 2.1. Characterization of the Manufactured Materials

Figure 1 shows the X-ray diffractograms of pure ZnO NCs and $V_2O_5$-ZnO nanostructured materials with four $V_2O_5$ NC proportions. Diffraction peaks at 2θ of 31.71°, 34.29°, 36.15°, 47.61°, 56.53°, 62.81°, 67.85°, and 69.14° are distinguished in the X-ray diffractograms of pure ZnO sample that could be accredited to (100), (002), (101), (102), (110), (103), (112), and (201) crystal planes, respectively, affirming the inclusion of hexagonal wurtzite ZnO (JCPDS 36-1451) [22,23]. Obviously, all the mesoporous $V_2O_5$-ZnO nanostructured materials acquired diffraction patterns identical to those of pure ZnO, demonstrating the existence of crystal planes of hexagonal wurtzite ZnO in such samples. The presence of polycrystalline NCs was confirmed by the observation of narrow and sharp peaks in the diffractograms of all the investigated samples. In the diffractograms of the prepared nanocomposites, there were no diffraction peaks of $V_2O_5$ phase, presumably due to their extensively scattered nature and tiny particle sizes, as well as their low concentration. In addition, there were no further peaks in the diffractograms of the synthesized $V_2O_5$-ZnO nanostructured materials, showing that high purity nanocomposites were produced.

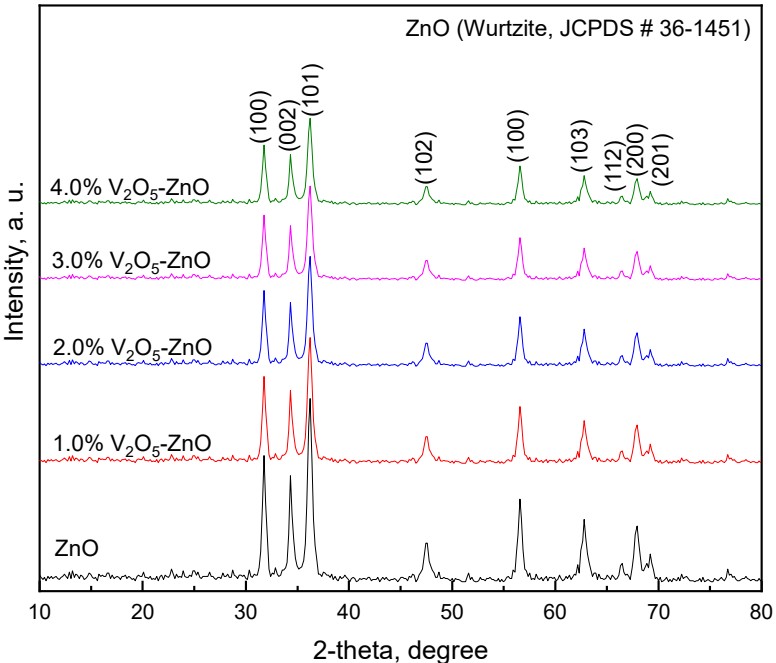

**Figure 1.** XRD patterns of pure ZnO sample and $V_2O_5$-ZnO nanostructured materials with various $V_2O_5$ concentrations.

It is extremely difficult to strike a balance between good crystallinity and a vast surface area of heterogeneous photocatalysts, such that they are frequently used in combined forms to provide high stability and a large number of accessible sites. In this context, Table 1 summarizes the findings of $N_2$ physisorption, which is a prospective technique for evaluating the specific surface area, pore size distribution, and other surface features of the different composites. Figure 2 shows $N_2$ adsorption/desorption isotherms and pore size distribution for the samples made of pure ZnO NCs and the 3.0 wt.% $V_2O_5$-ZnO nanocomposite. All the obtained isotherms seemed to have isotherms of type IV accompanied by hysteresis loops of H3 type (IUPAC), suggesting the inclusion of mesoporous texture within the manufactured samples [19,22]. Furthermore, the presence of homogeneous cylindrical channels, which characterize the mesoporous structure, was confirmed by the detection of capillary condensation in the relative pressure area (0.42–0.92) (Figure 2). Table 1 demonstrates that the surface area of the studied nanocomposites marginally fell as the proportion of $V_2O_5$ increased. It is clear that ZnO NCs had a relatively large surface area (145 $m^2$ $g^{-1}$), which

decreased to 118 m$^2$ g$^{-1}$ by applying a 4.0 wt.% V$_2$O$_5$-ZnO nanocomposite because some ZnO pores were blocked by the V$_2$O$_5$ NCs. The prevalence of mesoporous texture within pure ZnO NCs and V$_2$O$_5$-ZnO nanocomposite samples was proven by the narrow and sharp pore size curves for such samples (Figure 2 inset). It was also revealed from the data in Figure 2 inset that, as the proportion of V$_2$O$_5$ increased from 1 to 3.0 wt.%, the mean pore diameter of pure ZnO NCs reduced from 9.8 to 9.4 nm.

**Table 1.** Surface area, bandgap, and photooxidative desulfurization rate of thiophene using ZnO and V$_2$O$_5$-ZnO photocatalysts.

| Samples | S$_{BET}$ m$^2$ g$^{-1}$ | Bandgap, eV | k, min$^{-1}$ | R$^2$ | r, µmolL$^{-1}$ min$^{-1}$ |
|---|---|---|---|---|---|
| ZnO | 145 | 3.21 | $5.4 \times 10^{-4}$ | 0.990 | 3.877 |
| 1.0 % V$_2$O$_5$-ZnO | 130 | 3.02 | 0.00248 | 0.992 | 17.684 |
| 2.0% V$_2$O$_5$-ZnO | 124 | 2.95 | 0.00801 | 0.994 | 57.119 |
| 3.0% V$_2$O$_5$-ZnO | 119 | 2.90 | 0.0166 | 0.994 | 118.374 |
| 4.0% V$_2$O$_5$-ZnO | 118 | 2.89 | 0.0173 | 0.992 | 123.366 |

S$_{BET}$: surface area, k: reaction rate constant, r: desulfurization rate.

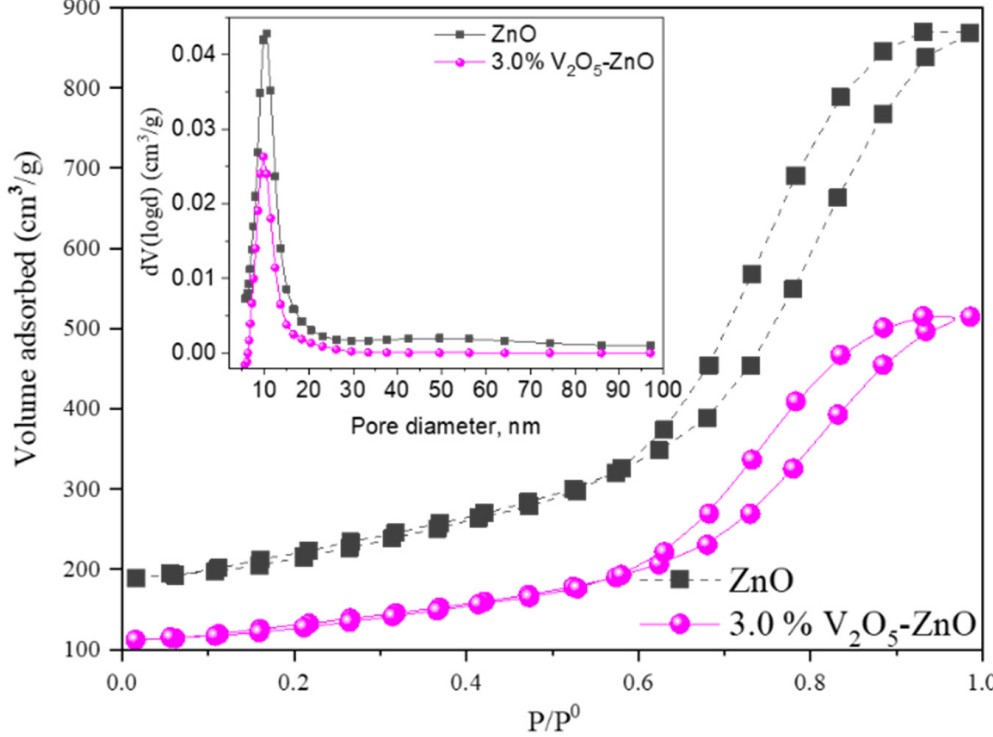

**Figure 2.** Nitrogen sorption isotherms of pure ZnO NCs and 3 wt.% V2O5-ZnO nanocomposites; inset shows pore size distribution curves for pure ZnO NCs and 3 wt.% V2O5-ZnO nanocomposite.

The microstructure of pure ZnO NCs and 3.0 wt.% V$_2$O$_5$-ZnO heterojunction samples were investigated using transmission electron microscopy images. Two-dimensional (2D) hexagonal ZnO NCs with a mean diameter of 15 nm appeared to be extensively dispersed and fully homogeneous in size, as seen in the TEM image of pure mesoporous ZnO in Figure 3a. In contrast, a TEM image of 3 wt.% V$_2$O$_5$-ZnO nanocomposite (Figure 3b) showed smaller NCs with sizes in the 4–6 nm range, having spherical shapes evenly dispersed throughout the ZnO surface. The HR-TEM of a selected area is seen in Figure 3c; the image highlights that both ZnO and V$_2$O$_5$ NCs were partly connected. Furthermore, it exhibits the typical lattice distances of 0.340 and 0.240 nm that could be accredited to the (110) and (101) crystal planes of V$_2$O$_5$ and ZnO, respectively [36,46].

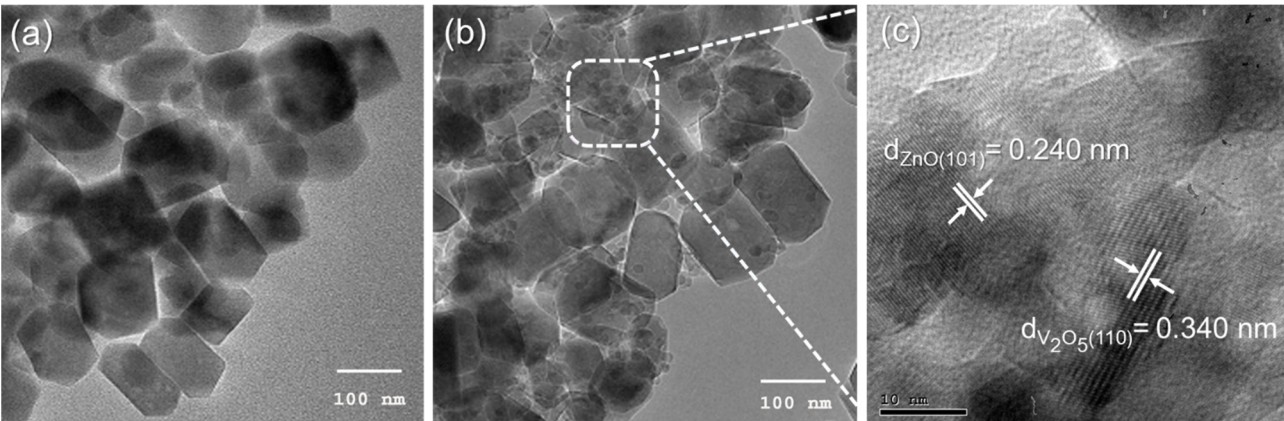

**Figure 3.** TEM photographs of (**a**) pure ZnO, (**b**) 3 wt.% $V_2O_5$-ZnO nanocomposite, and (**c**) HR-TEM photograph of 3 wt.% $V_2O_5$-ZnO nanocomposite.

The chemical states of 3.0 wt.% $V_2O_5$-ZnO nanocomposite was analyzed through X-ray photoelectron spectroscopy (XPS). Only V, Zn, and O were detected from the data of the XPS spectral curves of the $V_2O_5$-ZnO nanocomposite, as shown in Figure 4. The presence of two peaks at 524.6 and 517.1 eV in the XPS graph of V 2p might be ascribed to V $2p_{1/2}$ and V $2p_{3/2}$, correspondingly. As a result, it was possible to confirm the existence of $V^{5+}$ inside the manufactured nanocomposite (Figure 4a) [47–50]. Figure 4b shows two obvious peaks for Zn $2p_{3/2}$ and Zn $2p_{1/2}$ situated at the binding energy values of 1021.63 and 1044.68 eV, respectively, which belong to the $Zn^{2+}$ state [51–53]. The binding energy difference between Zn $2p_{3/2}$ and Zn $2p_{1/2}$ was determined to be 23.0 eV, suggesting the presence of a $Zn^{2+}$ species that fits the Zn-O bonds [53]. For the $V_2O_5$-ZnO nanocomposite, the O 1s peak was divided into three peaks at 529.2, 530.7, and 532 eV, as shown in Figure 4c. The inclusion of lattice oxygen throughout $V_2O_5$-ZnO was attributed to the peaks observed at the binding energy values of 529.2 and 530.7 eV, correspondingly [54]. The signal at 532 eV, on the other hand, might be attributed to the existence of oxygen species connected to the created nanocomposite in the form of hydroxyl groups, which could be generated when water molecules are adsorbed on the nanocomposite surface [55].

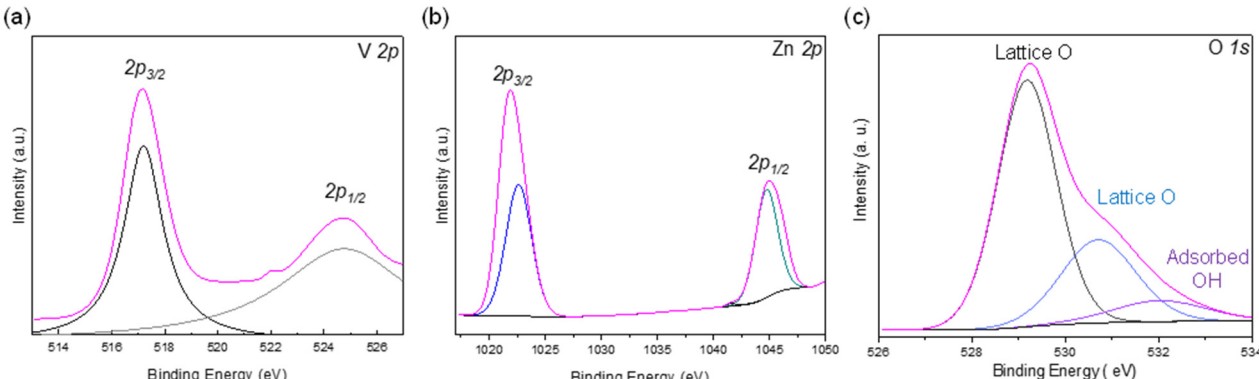

**Figure 4.** XPS examination of 3 wt.% $V_2O_5$-ZnO nanocomposite progressed by (**a**) V 2p, (**b**) Zn 2p, and (**c**) O 1s emissions.

UV-Vis spectroscopy was employed to evaluate the optical characteristics of the produced pure ZnO and $V_2O_5$-ZnO nanostructured materials. The absorption edge of pure ZnO was observed at 386 nm, whereas those of 1.0, 2.0, 3.0, and 4.0 wt.% $V_2O_5$-ZnO nanocomposites were approximated at 410, 420, 428, and 429 nm, correspondingly, as shown in Figure 5. The photocatalysts' absorption spectra demonstrated intensive light harvesting in a comprehensive, visible portion of the spectrum (200–800 nm), demonstrating

their exceptional photonic effectiveness. It is interesting to note that, by integrating $V_2O_5$ with ZnO, visible-light responses throughout $V_2O_5$-ZnO nanostructured materials are significantly improved. The bandgap energy rates of the manufactured $V_2O_5$-ZnO nanostructured materials were computed using the mathematical equation $(\alpha h\nu)^{1/n} = A(h\nu - E_g)$, where $\alpha$ refers to the absorption coefficient, h refers to the Planck constant, $\nu$ expresses the frequency, n = 2 for the indirect transition, A is a constant, and $E_g$ represents the bandgap; the data are displayed in Figure 5b. The results shown in Figure 5b disclosed that the bandgap energy for pure ZnO NCs was approximated at 3.21 eV, whereas the bandgap energy values decreased in the sequence of 3.02, 2.95, 2.90, and 2.89 eV for 1.0, 2.0, 3.0, and 4.0 wt.% $V_2O_5$-ZnO nanostructured materials. Consequently, the application of $V_2O_5$ in ZnO was considered to be necessary for achieving declined bandgap values in the manufactured nanostructured materials. $V_2O_5$-ZnO nanocomposites were found to acquire significantly better optical properties, allowing for more efficient utilization of solar light.

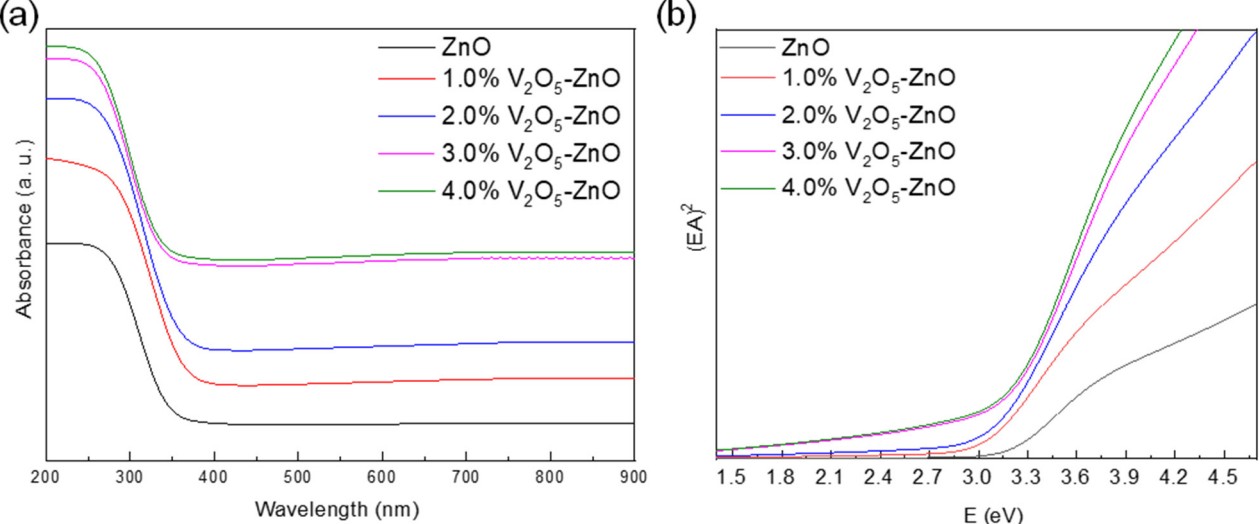

**Figure 5.** (**a**) DRS spectral graphs of all manufactured samples; (**b**) graph of transferred Kubelka–Munk vs. adsorbed light energy for pure ZnO and $V_2O_5$-ZnO nanocomposites.

## 2.2. Photocatalytic Performance Assessment

By measuring the TPOD during visible irradiation, the photocatalytic activities of the produced pure ZnO NCs and $V_2O_5$-ZnO nanostructured materials at varying $V_2O_5$ fractions were examined. The desulfurization degree measured in the absence of the photocatalyst clarified that the quantity of thiophene did not alter after 3 h of visible irradiation. Furthermore, in the dark, thiophene adsorption supporting all $V_2O_5$-ZnO nanostructured materials was insignificant. Figure 6a depicts the relationship between the degree of thiophene desulfurization and illumination time after visible irradiation. The findings demonstrated that, as the irradiation period increased, the desulfurization degree rapidly improved. After 150 min of irradiation, the desulfurization percentage employing pure ZnO NCs was indeed found to be around 7%. On the other hand, the desulfurization degree was greatly enhanced by increasing the $V_2O_5$ percentages throughout the manufactured $V_2O_5$-ZnO nanostructured materials, reaching 29, 70, 93, and 94% by applying 1.0, 2.0, 3.0 and 4.0 wt.% $V_2O_5$ NCs, respectively. The variance in desulfurization extent was minimal when $V_2O_5$ percentages rise exceeded 3 wt.%. As a result, the optimal 3.0 wt.% $V_2O_5$-ZnO nanocomposite was nearly able to fully oxidize thiophene after 150 min, which was 13.28 times significantly greater than that of pure ZnO NCs. The outcomes of Table 1 elucidated that the rate of thiophene desulfurization by applying ZnO NCs was assessed to be 3.87784 $molL^{-1}$ $min^{-1}$, while those of 1.0, 2.0, 3.0 and 4.0 wt.% $V_2O_5$-ZnO nanocomposites were in the sequence of 17.684, 57.119, 118.374 and 123.366 $molL^{-1}$ $min^{-1}$ (Table 1). Intriguingly, the thiophene desulfurization rate by ap-

plying the 3.0 wt.% $V_2O_5$-ZnO nanocomposite was found to be 30.5 times higher than that of ZnO NCs. Actually, thiophene's desulfurization performance when applying the synthesized materials was much superior to that of the earlier described nanomaterials [25]. The kinetic studies of the photocatalytic oxidative desulfurization of thiophene were carried out; the observed linear relationship of this photocatalytic process might be considered to be a pseudo-first-order style, known as the Langmuir-Hinshelwood model [56,57]. The Langmuir-Hinshelwood model is well-suited to heterogeneous photocatalysis, as seen in the following formula: rate of desulfurization ($r$) = $-dC/dt$ = k$C$.

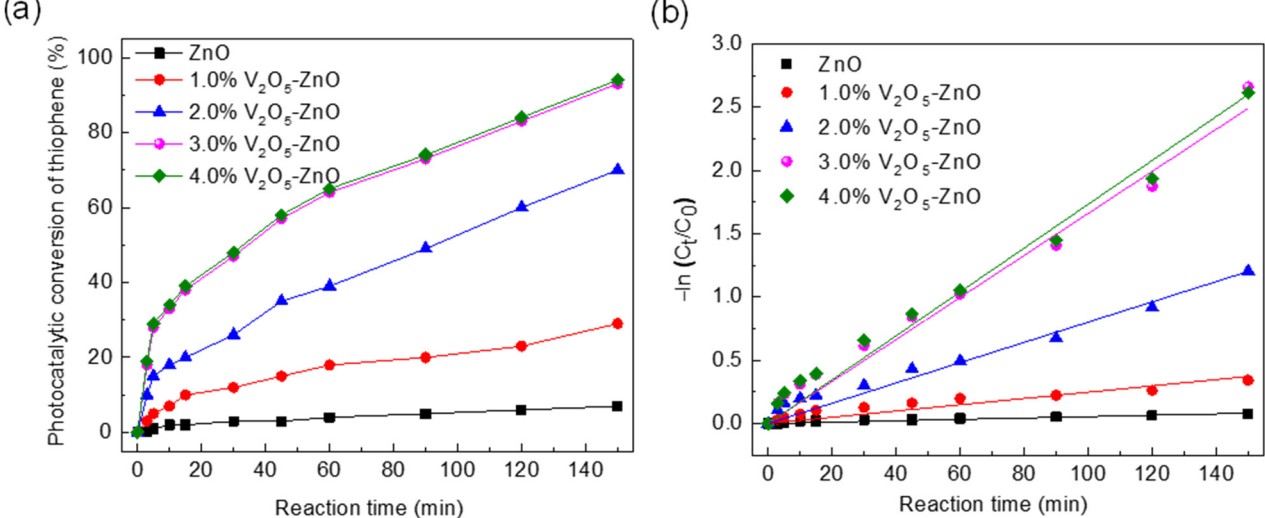

**Figure 6.** (**a**) The graphical representation of $-\ln (C_t/C_o)$ vs. visible irradiation duration when applying pure ZnO and $V_2O_5$-ZnO nanocomposites; (**b**) the thiophene photocatalytic transformation when applying pure ZnO and $V_2O_5$-ZnO nanostructured materials under visible illumination.

The reaction's velocity constant (k) might be estimated using the relationship involving $-\ln(C_t/C_o)$ and illumination time (t), where $C_o$ and $C_t$ are the thiophene amounts at the start of the irradiation and at different irradiation time durations, respectively. The $-\ln(C_t/C_o)$ vs. time relationship is shown in Figure 6b, and Table 1 shows the resulting k values. The velocity constant value for ZnO NCs was assessed to be $5.4 \times 10^{-4}$ min$^{-1}$, whereas those for 1.0, 2.0, 3.0, and 4.0 wt.% $V_2O_5$-ZnO nanocomposites were found in the sequence of 0.00248, 0.00801, 0.0166, and 0.0173 min$^{-1}$ (Table 1). The velocity constant while using the 3.0 wt.% $V_2O_5$-ZnO nanocomposite was 30.7 times larger than when using pure ZnO NCs, according to the results in Table 1.

The impact of the photocatalyst's dosage on the thiophene desulfurization level was investigated. Figure 7a reflects the thiophene desulfurization extent at different applied loads of the 3.0 wt.% $V_2O_5$-ZnO photocatalyst. When the catalyst dose increased from 0.4 to 1.6 g/L, the desulfurization extent increased from 75% to 100%; at such a dose (1.6 g/L), the desulfurization degree was verified to reach the maximum (100%) after 90 min of visible irradiation that might be attributed to the increased ·OH radical content. Nevertheless, when the $V_2O_5$-ZnO nanocomposite dosage surged above 1.6 g/L, the turbidity of the fluid in the system increased, hindering light propagation [58]. Due to excessive light scattering, the total degree of desulfurization was reduced to 85% when even a large dosage (2.0 g/L) of $V_2O_5$-ZnO nanocomposite was used [59].

Durability and stability are critical parameters regarding the application of recycled photocatalysts in commercial implementation. During the TPOD process using the 3.0 wt.% $V_2O_5$-ZnO photocatalyst, the durability and reusability of this material were examined for five runs. After the first and second runs, the photocatalytic performance was substantially retained, almost unchanged, but the effectiveness lowered to 99% in the next three runs, as shown in Figure 7b. The photocatalyst's durability and stability were affirmed using

the XRD measurements of the 3.0 wt.% $V_2O_5$-ZnO photocatalyst before and after the photocatalytic processes throughout 10 h irradiation, as shown in Supplementary Figure S1. The attained outcomes revealed that the manufactured $V_2O_5$-ZnO nanocomposites were extremely durable and stable, with no crystallinity collapse.

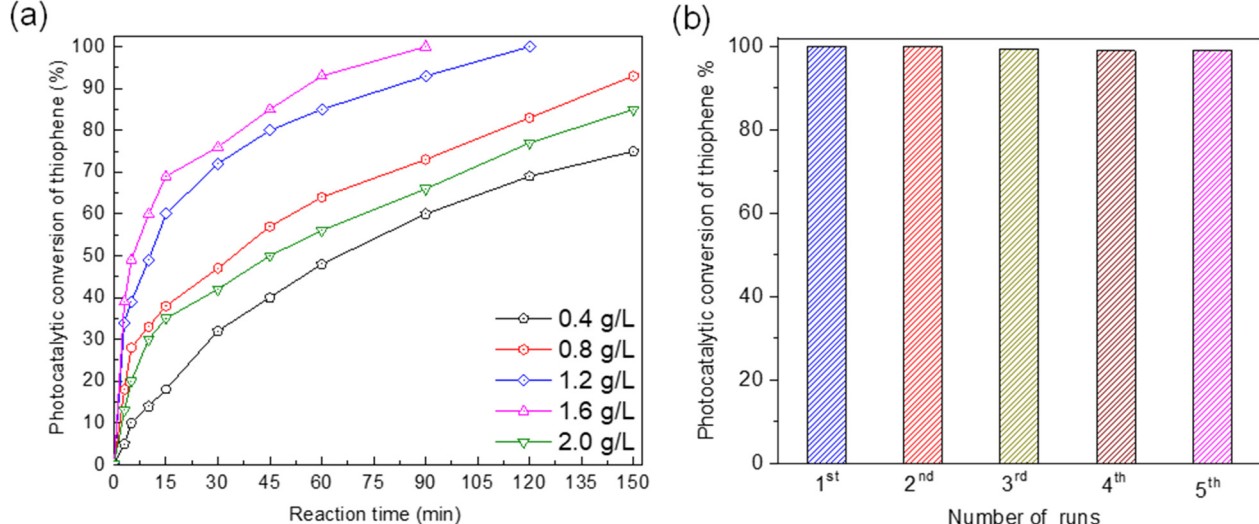

**Figure 7.** (**a**) Consequence of 3 wt.% $V_2O_5$-ZnO nanocomposite dosage amount on thiophene photocatalytic transformation; (**b**) recyclability of 3 wt.% $V_2O_5$-ZnO nanocomposite for 5 intervals under visible illumination.

To support the attained findings, PL spectra were employed to examine the photoinduced charge carriers' recombination rate. It is well-known that an increase in the separation amid the photogenerated charges suppresses the recombination rate between such charge carriers, resulting in a low PL intensity. The PL spectral graphs of $V_2O_5$-ZnO heterojunctions with various $V_2O_5$ percentages that were triggered at 365 nm are illustrated in Figure 8a. In the $V_2O_5$-ZnO nanostructured materials, the PL intensity observed at 452 nm gradually decreased with the increase in $V_2O_5$ proportions (from 0.0 up to 4.0 wt.%) and slightly red-shifted to relatively large wavelengths, suggesting that the progression of such nanocomposites can inhibit the electron–hole pair recombination and potentially lead to lower PL intensity. Thus the resulting heterojunctions acquired enhanced photonic effectiveness.

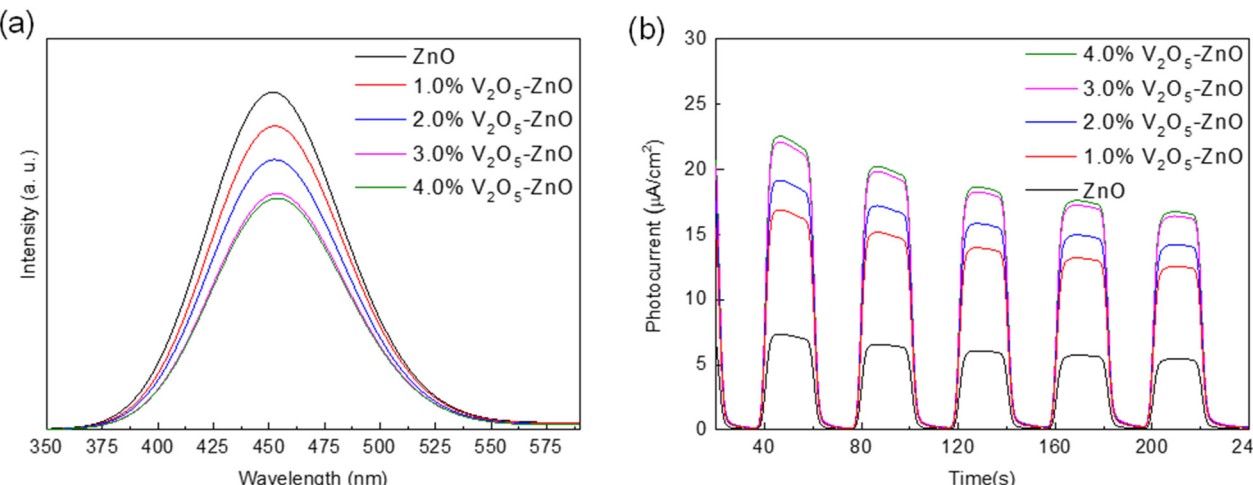

**Figure 8.** (**a**) PL spectral curves and (**b**) photoexcited electrons' transmission efficiency when applying all manufactured materials.

Photocurrent assessments, on the other hand, were used to confirm the photoinduced carriers' recombination rate. The transient photocurrent assessments of mesoporous $V_2O_5$-ZnO nanocomposites when applied as photoanodes were investigated. The transient responses of electrodes produced with mesoporous $V_2O_5$-ZnO nanostructured materials comparable to pure ZnO NCs are shown in Figure 8b. Both the dark current and photocurrent for $V_2O_5$-ZnO nanostructured materials may quickly attain the equilibrium state via on/off irradiation, which is consistent with the previous reports [60,61]. Remarkably, the photocurrent response of the 3 wt.% $V_2O_5$-ZnO heterojunction reached three folds (22.7 µA cm$^{-2}$) that of the pure ZnO (7.2 µA cm$^{-2}$), showing that the $V_2O_5$-ZnO nanocomposites improved photoinduced carrier separation, which surely positively affects the photocatalytic efficiency. The outcomes of the PL and photocurrent responses were found to be compatible with those of the photocatalytic efficiency assessment.

To ensure that the TPOD process was completed, The gases yielded from the reaction were trapped in NaOH (0.2 M), and then $Ba(NO_3)_2$ (0.2 M) was added to the solution. The obtained precipitate was separated and dried before being identified by XRD analysis [62]. As shown in Supplementary Figure S3, the results of the XRD diffractogram of the produced precipitate confirmed the production of barium carbonate (ICDD-PDF No. 05-0378). This can be accredited to thiophene oxidation into $CO_2$, which dissolves in sodium hydroxide solution and reacts with $Ba(NO_3)_2$ to produce $BaCO_3$. Remarkably, $HNO_3$ (aq) was used to dissolve $BaCO_3$ in the white precipitate, and there was some white precipitate left over that did not dissolve. XRD analysis was performed to analyze the residual precipitate. According to the XRD data, the precipitate was identified as barium sulfate, which is consistent with ICDD-PDF No. 24-1035 (Supplementary Figure S4). This implied that thiophene was also oxidized to SO3. Therefore, the photocatalytic oxidation of thiophene often results in the production of $SO_3$ and $CO_2$ gaseous products [62].

The photocatalytic efficacy of the $V_2O_5$-ZnO nanostructured materials outperformed that of the pure ZnO NCs due to their major benefits of the different electronic structures of both $V_2O_5$ and ZnO (Figure 9). The photocatalytic efficiency of the $V_2O_5$-ZnO nanocomposite with 3.0 wt.% $V_2O_5$ outperformed that of the pure ZnO NCs. The influence of the $V_2O_5$-ZnO heterojunction's theoretical band structure on photocatalytic activities was investigated. The bandgap energy rates of both semiconductors ($V_2O_5$ and ZnO) were initially measured, and the alignment locations of such semiconductors were then calculated as follows [63]:

$$E_{CB} = X - E_e - 0.5E_g \tag{1}$$

$$E_{VB} = E_{CB} + E_g \tag{2}$$

where $E_{VB}$ refers to the valence band's (VB) edge potential, $E_{CB}$ refers to the conduction band's (CB) edge potential, $E_g$ represents the semiconductor's bandgap energy, $X$ refers to semiconductor electronegativity ($X_{V2O5}$ = 6.10 eV and $X_{ZnO}$ = 5.79 eV) [63], and $E_e$ refers to the free-electron energy on a hydrogen scale (4.5 eV). On the basis of the above equations, $V_2O_5$ acquired $E_{VB}$ of +2.78 and $E_{CB}$ of +0.43 eV, whereas ZnO acquired $E_{VB}$ of +2.89 eV and $E_{CB}$ of −0.31 eV, as seen in Figure 9a. Consequently, the resulting $V_2O_5$-ZnO heterojunction formation had a band structure that can be postulated [63]. Because of their close locations between their Fermi levels and band locations, both $V_2O_5$ and ZnO are perfect for creating junctions upon contact (Figure 9b). The photocatalytic performance improvement in the $V_2O_5$-ZnO heterojunction might be attributed to the simulated junction design and probably to the aligned bands of $V_2O_5$-ZnO heterojunctions that are revealed [53,61,64]. Owing to the electric field created at the heterojunction of the $V_2O_5$-ZnO combination, the photogenerated electrons can pass from the CB of the aligned, visible-light active $V_2O_5$ to the CB of visible-light inactive ZnO, leaving the generated holes remaining at the CB of $V_2O_5$. Thus, Figure 9b suggests a probable pathway for TPOD during visible irradiation. In other words, the charge carriers were separated when a $V_2O_5$-ZnO heterojunction was irradiated with an energy rate greater than or equal to the bandgap energy values of both $V_2O_5$ and ZnO. The photogenerated electrons easily flowed from the CB of $V_2O_5$

to that of ZnO. The photoinduced electrons reacted with $O_2$ and $OH^-$, resulting in the formation of $O_2^{·-}$, OOH, and $OH^-$. Furthermore, the photoinduced holes trapped by $OH^-$ resulted in the formation of ·OH radicals, as revealed by supplementary experiments for radical confirmation (Figure S5). As a result of the complete oxidation, TPOD could be achieved on both sides of the heterojunction, releasing both $CO_2$ and $SO_3$ gaseous products. Evidently, the retarded recombination rate of photogenerated carriers was successfully accomplished on the interface of $V_2O_5$-ZnO heterojunction throughout the photocatalytic oxidation of thiophene, and then the created radicals such as ·OOH, ·OH, and $O_2^{·-}$ as oxidizing agents could complete the TPOD process to $CO_2$ and $SO_3$ gaseous products. On the other hand, the photogenerated electrons were retrained by $O_2$ to generate $O_2^{·-}$. Subsequently, the acquired $O_2^{·-}$ transformed to $H_2O_2$, which was then broken down into the ·OH radicals. As a result, the produced mesoporous $V_2O_5$-ZnO heterojunctions may minimize recombination and increase photogenerated charges' separation. Thus, by synthesizing $V_2O_5$ NCs on a mesoporous ZnO surface, a nanojunction arrangement could be created, enhancing photocatalytic efficiency for full thiophene oxidation to $SO_3$ and $CO_2$.

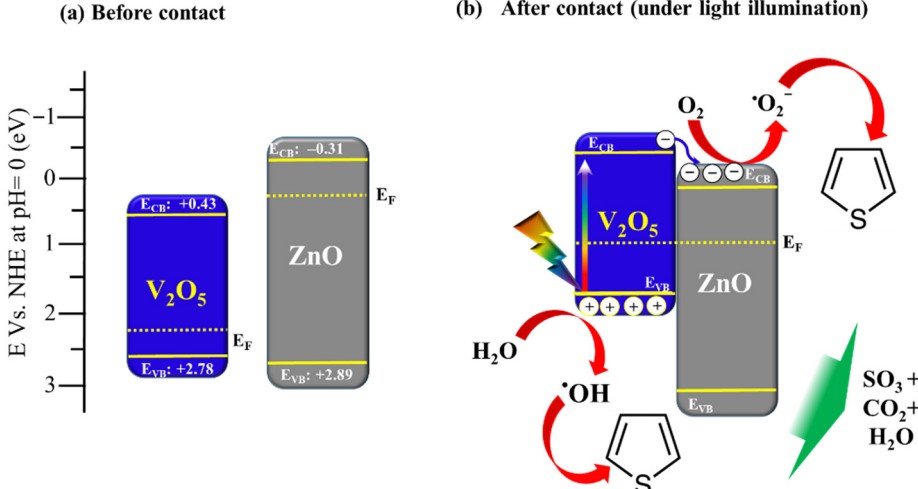

**Figure 9.** Predicted band positions (**a**) before contact of ZnO and $V_2O_5$ and the proposed photocatalytic pathway of $V_2O_5$-ZnO nanostructured composites after contact (**b**) for TPOD under visible irradiation.

## 3. Experimental

### 3.1. Materials

Ammonium metavanadate ($NH_4VO_3$, 99%) and zinc nitrate hexahydrate ($Zn(NO_3)_2 \cdot 6H_2O$, 99 %) were utilized as essential chemicals for nanocomposite manufacturing. The sulfur-containing compounds were modeled using thiophene ($C_4H_4S$). A Pluronic P-123 block copolymer surfactant was employed as a template. As reagents, absolute ethanol ($C_2H_5OH$, 99.8%), acetic acid ($CH_3COOH$, 99.8%), and hydrochloric acid (HCl, 37%) were adopted. Acetonitrile ($CH_3CN$, 99.8%) was applied as a solvent for thiophene. All the chemicals and reagents were outfitted from Sigma-Aldrich (Darmstadt, Germany).

### 3.2. Construction of Mesoporous $V_2O_5$-ZnO Nanostructured Materials

The Pluronic P-123 block copolymer was utilized as a template for constructing ZnO photocatalysts with a simple routine. The molar proportions of ZnO, P-123 block copolymer, absolute ethanol, hydrochloric acid, and acetic acid were applied with ratios of 1.00:0.02:50.00:2.25:3.75, correspondingly. Then, 0.2 g of the P-123 block copolymer was frequently mixed with absolute ethanol (30 mL) along with 1 h of vigorous agitation. The preceding combination was then mixed with zinc nitrate hexahydrate, acetic acid, and hydrochloric acid with quantities of 12.4 g, 2.3 mL, and 3.5 mL, correspondingly. To initiate the polymerization of both zinc ions and the P-123 block copolymer, the preceding combi-

nation was dried at 40 °C and about 60% relative humidity for 12 h. The resulting material was dried at 65 °C for 12 h. To obtain the mesoporous ZnO photocatalyst, the final product was sintered at 550 °C for 4 h.

$V_2O_5$ NCs were deposited into the mesoporous ZnO photocatalyst at varied $V_2O_5$ percentages (1.0, 2.0, 3.0, and 4.0 wt.%) and subsequently calcined for 3 h at 450 °C. Mesoporous ZnO (1.0 g) was typically mixed with ethanol (100 mL) and ultrasonicated for 5 min. After that, ammonium metavanadate (2.0 g) was added to the preceding suspension solution, which was agitated for 1 h. The solution was then dried by evaporating ethanol and placed in the furnace at 110 °C overnight. After that, the powdered material was heated for 3 h at 450 °C to obtain the $V_2O_5$-ZnO heterojunctions with 1.0, 2.0, 3.0, and 4.0 wt.% $V_2O_5$.

### 3.3. Characterization of the Manufactured Materials

A JEOL JEM-2100F transmission electron microscope (TEM, Tokyo, Japan) working at 200 kV was used to capture images of the manufactured $V_2O_5$-ZnO samples dispersed over carbon-coated grids. Utilizing a D4 Endeavour X diffractometer (Bruker AXS, Billerica, MA, USA) configured with Cu K radiation (=1.5418 Å), XRD measurements for $V_2O_5$-ZnO powders were explored. Using a Quantachrome NOVA 2000 surface area analyzer (Ostfildern, Germany), $N_2$ isotherms were obtained after degassing $V_2O_5$-ZnO materials in a vacuum for 10 h at 200 °C. The specific surface area, as well as pore size distribution of the produced $V_2O_5$/ZnO materials, were assessed from the resulting isotherms. UV-Vis diffuse reflectance spectroscopy (DRS) was measured employing a Shimadzu UV-3600 UV/Vis/NIR spectrophotometer (Kyoto, Tapan) and barium sulfate as a reflectance reference in ambiance. Employing Thermo Scientific K-ALPHA spectrometer (Waltham, MA, USA) fitted with monochromatic Al K$\alpha$ X-ray sources (1486.6 eV), the X-ray photoelectron spectroscopy (XPS) measurements of the 3.0 wt.% $V_2O_5$-ZnO samples were determined after calibration to the standard binding energy of the carbon peak located at 284.6 eV. The PL spectral curves of the $V_2O_5$-ZnO nanostructured materials were recorded using an RF-5301 PC spectrophotometer (Shimadzu, Kyoto, Tapan) and a xenon lamp (150 W, Ushio, Tokyo, Japan) as a source of excitation at a wavelength of 365 nm in ambiance. A three-electrode device was used to analyze the transient photocurrent spectra of the manufactured samples employing a Zahner Zennium electrochemical workstation (Kronach, Germany) in $Na_2SO_4$ (0.5 M).

### 3.4. Photocatalytic Performance Assessment

In a photoreactor, the photocatalytic desulfurization of thiophene was carried out in the existence of a steady supply of the oxidizing agent (oxygen gas). In the thiophene solution with acetonitrile as the solvent (sulfur content at the start = 600 ppm), a 1 g/L photocatalyst sample was dispersed. To reach thiophene adsorption equilibrium in the photocatalyst, the previous solution was stirred for 30 min in the dark. The temperature of the combination was sustained at around $15 \pm 2$ °C via cold water circulation. Visible illumination originated from a 300 W xenon lamp with a 45 mW cm$^{-2}$ intensity and a wavelength range of 400–700 nm, which was enclosed in the reaction vessel. For 150 min, the system was exposed to visible illumination. A sample of the reaction mixture was taken at specific illumination durations and centrifuged to extract the solid photocatalyst. The concentration of thiophene in the sample was then determined using GC-FPD (Agilent 7890, FFAP column, Santa Clara, CA, USA).

### 4. Conclusions

Under visible-light exposure at an ambient temperature, $V_2O_5$-ZnO nanocomposites were effectively produced using a simple sol–gel routine for thiophene photocatalytic oxidative desulfurization. The photocatalytic effectiveness of the 3 wt.% $V_2O_5$-ZnO heterojunction was greater, with a velocity constant value of 0.0166 min$^{-1}$, which was 30.7-fold higher than that of pure ZnO NCs (0.00054 min$^{-1}$). The development of a nanojunction

arrangement using $V_2O_5$ NCs on a mesoporous ZnO surface might improve photocatalytic efficiency for perfect thiophene oxidation to $SO_3$ and $CO_2$ gases. The synergistic influence, which encompasses quick thiophene transportation, a lesser light scattering action, and the generation of a substantial amount of ·OH radical content, might bring about a pronounced degree of thiophene desulfurization. The mesoporous $V_2O_5$-ZnO nanocomposites created using this simple method showed promise in TPOD processes using solar energy.

**Supplementary Materials:** The following supporting information can be downloaded at: https://www.mdpi.com/article/10.3390/catal12090933/s1, Figure S1: UV-Vis-NIR spectrum of pure $V_2O_5$ and estimated bandgap in the inset; Figure S2: XRD patterns of the 3 wt.% $V_2O_5$/ZnO photocatalyst before and after photocatalytic reactions; Figure S3: XRD patterns of the produced $BaCO_3$; Figure S4: XRD patterns of the produced $BaSO_4$; Figure S5: Photocatalytic oxidative desulfurization of thiophene over the optimized nanocompo-sites using different radical scavengers.

**Author Contributions:** Conceptualization, A.S. and M.A.; formal analysis, A.S. and M.A.; funding acquisition, Z.I.Z.; methodology, M.A.; project administration, Z.I.Z.; software, Z.I.Z.; supervision, A.S.; validation, A.S.; visualization, A.S. and M.A.; writing—original draft preparation, A.S.; writing—review and editing, A.S. All authors have read and agreed to the published version of the manuscript.

**Funding:** Taif University Researchers Supporting Project (TURSP-2020/42), Taif University, Taif Saudi Arabia.

**Data Availability Statement:** All data that support the findings of this study are included within the article (and Supplementary Materials).

**Acknowledgments:** The authors thank the Taif University Researchers Supporting Project (TURSP-2020/42), Taif University, Taif Saudi Arabia. In the meantime, A. Shawky appreciates the technical support offered by his institution CMRDI in Egypt.

**Conflicts of Interest:** The authors declare no conflict of interest.

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
