# Peer review of "Photocatalytic Oxidative Desulfurization of Thiophene by Exploiting a Mesoporous V2O5-ZnO Nanocomposite as an Effective Photocatalyst"

_catalysts, doi:10.3390/catal12090933_

Round 1
Reviewer 1 Report
This study used a simple method to prepare V2O5/ZnO nanocomposites for thiophene photocatalytic oxidative desulfurization (TPOD) under visible light. This research is fascinating. However, some key issues should be addressed. So I think a major revision is needed.
1. In the abstract section, "Over a 3.0 wt. % V2O5/ZnO ... faster than pure ZnO NPs" has the same meaning as the previous sentence, and it is recommended to delete it.
2. Why does the presence of narrow and sharp peaks in the XRD spectrum indicate the presence of polycrystalline nanoparticles?
3. I think the type of the sample hysteresis loop is H3, and the author is advised to check it carefully and consider whether to modify it.
4. In the introduction, the author mentioned the preparation of 2D V2O5/ZnO nanocomposites but described V2O5 and ZnO as nanoparticles. The logic is confusing before and after, and please modify the author. To explain the material's morphology more clearly, the authors need to supplement the SEM images of ZnO and 3 wt% V2O5/ZnO and the corresponding mapping images.
5. In Fig. 3c, the corresponding lattice fringes of V2O5 cannot be confirmed, and the authors need to add more substantial evidence that V2O5/ZnO composites were prepared.
6. From the DRS data, pure ZnO cannot be excited by the experimental light (400-700 nm). Therefore, the final photocatalytic mechanism is not rigorous. The author should provide pure V2O5 DRS data and estimate its Eg and the corresponding valence and conduction band positions.
7. Authors need to supplement active species capture experiments. Please refer to Liu C, Mao S, Shi M, et al. Chemical Engineering Journal, 2022, 449: 137757.
8. In lines 368-370, the valence and conduction band potentials of ZnO and V2O5 described by the author do not match the potential values depicted in Fig. 9. Why?
9. Suggestion: More recent papers involving photocatalytic environmental remediation can be referenced in the manuscript. (Liu C, Mao S, Wang H, et al. Chemical Engineering Journal, 2022, 430: 132806; Liu C, Mao S, Shi M, et al. Journal of Hazardous Materials, 2021, 420: 126613).
Author Response
"Please see the attachment."

Reviewer 2 Report
August, 17, 2022
The manuscript titled “Photocatalytic oxidative desulfurization of thiophene exploiting mesoporous V2O5/ZnO nanocomposite as an effective photocatalyst"
In this manuscript, authors reported synthesis of two-dimensional mesoporous V2O5/ZnO nanocomposites using a simple surfactant-assisted sol-gel method for thiophene photocatalytic oxidative desulfurization (TPOD) at ambient temperature. The results of this research are conveyed thoughtfully and completely, and they are consistent with the experimental findings. However, the authors failed to explain and draw out the novelty of the work, this aspect needs to be improved. This work is worthwhile to be publish in this journal after major revision. The following issues should be addressed:
1. Introduction is well-organized and well-written, but the importance and novelty of the research should be highlighted and more clearly stated. The authors give some examples of works in the bibliography, but which is the advantage of their work in comparison with those works.
2. In section 2.3, the conditions used for all characterization techniques should be added.
3. After cycle stability of photocatalytic reaction, whether morphology and crystal structure of the catalysts were changes?
4. The comparison on photocatalytic activity of as-prepared samples with some typical ZnO based photocatalysts ever reported should be added.
5. The authors should measure the by-products that produced after complete photodegradation activity of Thiophene.
6. A more comprehensive background for photocatalytic and applications of different metal oxides as a photocatalysts should be illustrated in the introduction for a wider readership. For example, some papers, which describe photocatalysts should be cited: https://doi.org/10.1016/j.jtice.2021.08.034, https://doi.org/10.1016/j.jmrt.2022.03.067, https://doi.org/10.1021/acsomega.1c03735, https://doi.org/10.1007/s11144-021-02050-4, https://doi.org/10.1007/s10971-022-05755-7
11. The authors are responsible for the English, which should be polished throughout the manuscript to clear some minor typo/grammar errors.
Hence, I recommend it accepted for publication after some major revisions.
Author Response
"Please see the attachment."

Reviewer 3 Report
The authors in the paper titled “Photocatalytic oxidative desulfurization of thiophene exploiting mesoporous V2O5/ZnO nanocomposite as an effective photocatalyst” showed the effect of V2O5 on degeneration of thiophene, together with detail preparation and characterization of the nanocomposite. The approach is sustainable and effective, which should be also accomplished with more mechanism evidence. The involvement of ROS can be further proved by the addition of specific quenchers as in Photochem. Photobiol. Sci., 2020, 19, 996. The amount of the gas products should be reported in % yield to confirm if there is any other product during the reaction. The comparison of full wavelength light source and light sources with narrow wavelength should be perform to (1) understand the involvement of thiophene excitation; and (2) if the full wavelength light induced side reactions. Also, the absorption intensities or said the wavelength matching should be mentioned and considered when comparing the PL intensity. Overall, the manuscript would be consider published after revision.
Other comments:
Abstract: “The photocatalytic effectiveness of the 3.0 wt.% V2O5/ZnO photocatalyst was the greatest among all 19 other samples, with a rate constant of 0.0166 min-1 that was 30.7 significantly greater than that of 20 pure ZnO NPs (0.00054 min‒1 ).” and “Over a 3.0 wt. % V2O5/ZnO nanocomposite, thiophene desulfuriza- 21 tion speed was 30.7 times faster than pure ZnO NPs.” seem duplicated.
line 36: H2S
line 197-198: V2O5
Author Response
"Please see the attachment."

Round 2
Reviewer 1 Report
Authors addressed very well most of my comments; The paper could be considered for publication.
Reviewer 2 Report
The manuscript is acceptable in the present form since authors have well addressed the questions proposed by referees.
Reviewer 3 Report
Thank you for the revised manuscript, which answers my question properly. I suggest accepting the current version for publication.